# Green Synthetic Approach: An Efficient Eco-Friendly Tool for Synthesis of Biologically Active Oxadiazole Derivatives

**DOI:** 10.3390/molecules26041163

**Published:** 2021-02-22

**Authors:** Bimal Krishna Banik, Biswa Mohan Sahoo, Bera Venkata Varaha Ravi Kumar, Krishna Chandra Panda, Jasma Jena, Manoj Kumar Mahapatra, Preetismita Borah

**Affiliations:** 1Department of Mathematics and Natural Sciences, College of Sciences and Human Studies, Prince Mohammad Bin Fahd University, Al Khobar 31952, Saudi Arabia; 2Roland Institute of Pharmaceutical Sciences, Biju Patnaik University of Technology Nodal Centre of Research, Berhampur 760010, India; ravikumar_bvv@yahoo.co.uk (B.V.V.R.K.); krishnachandrapanda@gmail.com (K.C.P.); jasma.jena@gmail.com (J.J.); 3Kanak Manjari Institute of Pharmaceutical Sciences, Rourkela 769015, India; manojbit07@gmail.com; 4CSIR-Central Scientific Instruments Organization, Chandigarh 160030, India; preetiboradu@gmail.com

**Keywords:** drug, green chemistry, microwave, oxadiazole, synthesis, biological activities

## Abstract

Green synthetic protocol refers to the development of processes for the sustainable production of chemicals and materials. For the synthesis of various biologically active compounds, energy-efficient and environmentally benign processes are applied, such as microwave irradiation technology, ultrasound-mediated synthesis, photo-catalysis (ultraviolet, visible and infrared irradiation), molecular sieving, grinding and milling techniques, etc. Thesemethods are considered sustainable technology and become valuable green protocol to synthesize new drug molecules as theyprovidenumerous benefits over conventional synthetic methods.Based on this concept, oxadiazole derivatives are synthesized under microwave irradiation technique to reduce the formation of byproduct so that the product yield can be increased quantitatively in less reaction time. Hence, the synthesis of drug molecules under microwave irradiation follows a green chemistry approach that employs a set of principles to minimize or remove the utilization and production of hazardous toxic materials during the design, manufacture and application of chemical substances.This approach plays a major role in controlling environmental pollution by utilizing safer solvents, catalysts, suitable reaction conditions and thereby increases the atom economy and energy efficiency. Oxadiazole is a five-membered heterocyclic compound that possesses one oxygen and two nitrogen atoms in the ring system.Oxadiazole moiety is drawing considerable interest for the development of new drug candidates with potential therapeutic activities including antibacterial, antifungal, antiviral, anticonvulsant, anticancer, antimalarial, antitubercular, anti-asthmatic, antidepressant, antidiabetic, antioxidant, antiparkinsonian, analgesic and antiinflammatory, etc. This review focuses on different synthetic approaches of oxadiazole derivatives under microwave heating method and study of their various biological activities.

## 1. Introduction

The green chemistry approach refers to the utilization of a set of principles that reduces the generation of chemical hazardous during the design, manufacture and use of chemical substances. This protocol plays a major role in controlling environmental pollution by using safer solvents, catalysts, suitable reaction conditions and thereby increases the atom economy and energy efficiency of the synthetic process. Hence, microwave-assisted synthesis followsthe green chemistry approach as it makes the synthetic process eco-friendly by reducing environmental pollution [1]. Microwave radiation energy offers significant benefits to carry out drug synthesis, including increased reaction rates, product yield enhancements, and cleaner reactions.The chemical transformations which take hours, or even days, to complete can now be completed in minutes with the help of microwave heating [2].

Similarly, the ultrasonic irradiation method is applied to accelerate various chemical reactions, including both homogeneous and heterogeneous systems. The use of ultrasound in organic synthesis involves specific activation based on the physical phenomenon, i.e., acoustic cavitations.In contrast, photo-catalytic reactions involve the use of ultra-violet, visible light and infrared radiation to generate new medicinally active compounds with diverse molecular structures. To carry out a photochemical reaction, the UV-visible spectra of the photoactive compounds are recorded. The photoactive compound is the molecule that can be electronically excited and undergoes chemical reaction from its excited state [3].

The grinding technique is also considered a green synthetic method to perform chemical reactions under solvent-free conditions with high product yield. Grinding of the recanting substances for a chemical reaction can be carried out by using mortar and pestle or by using a high-speed vibrating mill. Due to the collision between the reacting molecules, the chemical reaction is carried forward [4]. A milling technique like ball milling is considered to be one of the automated forms of mortar and pestle. In the case of theball mill, the reacting materials are placed in a reaction vessel equipped with grinding balls and covered with a lid. The vessel is allowed to shake at high-speed to carry out the chemical reactions [5].

Based on these above facts, oxadiazole derivatives are synthesized to reduce the formation of byproducts so that the product yield can be increased in less reaction time. Further, the structural motif like oxadiazole is drawing considerable attention for the development of new drug candidates with potential therapeutic activities including antibacterial, antifungal, antiviral, anticonvulsant, anticancer, antimalarial, antitubercular, anti-asthmatic, antidepressant, antidiabetic, antioxidant, antiparkinsonian, analgesic and antiinflammatory, etc.

Oxadiazole moiety is considered to be derived from furan ring by replacing two methane groups (-CH=) with two pyridine-type nitrogen (-N=).The aromaticity of oxadiazole is decreased due to the replacement of these groups in the furan ring so that it exhibits the property of conjugated diene. The oxadiazole ring is also recognized as furadiazole, furoxans, azoximes, oxybiazole, biozole and diazoxole. Oxadiazole derivatives are found to be a very weak base due to the inductive effect of the additional heteroatoms.Hence, there is the possibility of four isomers of oxadiazole, including 1,2,3-oxadiazole, 1,2,4-oxadiazole, 1,2,5-oxadiazole, 1,3,4-oxadiazole that depends on the position of nitrogen in the ring [6].

The therapeuticpotentials of oxadiazole derivatives mainly depend on the effective binding interactions of drug molecules with different receptors or enzymes in the biological systems and thereby eliciting diverse bioactivities. This review provides new insights into the rational design to develop potential oxadiazole-based medicinal agents with less toxicity and improved pharmacokinetic properties.

## 2. Green Chemistry Approaches

There are various green chemistry approaches to carry out different chemical reactions that include microwave irradiation (MWI), ultrasonication, photo-catalysis, grinding and milling methods (Figure 1). By applying these technologies, organic reactions become more efficient and economic by enhancing the rate of reaction with reduced reaction time and high product yield. Synthetic approaches like grinding or milling techniques involve the application of mechanochemistry for the rapid, clean, efficient and solvent-free synthesis of various biologically active compounds [7].

During the year 1991, the Environmental Protection Agency (EPA) and the National Science Foundation (NSF) initiated the Green Chemistry Program. P.T. Anastas and J.C. Warner have formulated twelve major principles of green chemistry to reduce or eliminate the risk of chemical hazards and environmental pollution [8,9,10,11].
Prevention of waste or byproducts: *It is essential to carry out the synthesis in such a way that the formation of waste or byproducts is less or absent.*Atom economy: *It represents the design of synthetic methods to maximize the incorporation of reactants (starting materials and reagents) to get the final products*.Use of less hazardous and toxic chemicals: *Various synthetic methods should be designed properly so that the use and generation of substances have less or no toxic effect on human health and the environment.*Designing of Safer chemicals: *The design of the chemical product should preserve efficacy while reducing toxicity.*Selection of Safer solvents: *Avoid the use of auxiliary materials (solvents, extractants) if possible, or otherwise, make them innocuous.*Energy efficiency: *Energy requirements should be minimized and conduct synthesis at ambient temperature and pressure.*Renewable feedstock: *Raw materials should be renewable.*Reduce derivatives: *Unnecessary derivatization should be avoided where possible.*Smart catalysis: *Selectively catalyzed processes are superior to stoichiometric processes.*Biodegradable design: *The design of chemical products should be in such a way that these can be degradable to innocuous products when disposed of*.Real-time analysis for pollution prevention: *Monitor the processes in real time to avoid excursions leading to the formation of hazardous substances.*Prevention of hazards and accidents: *Materials used in a chemical process should be selected to minimize hazardsand risk for chemical accidents.*

## 3. Chemistry of OxadiazoleMoiety

Oxadiazoles are five-membered heterocyclic compounds that possess one oxygen atom and two nitrogen atoms in the ring system [12]. Depending on the position of heteroatoms (oxygen or nitrogen), there are different isomeric forms of oxadiazole moiety such as 1,2,3-oxadiazole, 1,2,4-oxadiazole, 1,2,5-oxadiazole, 1,3,4-oxadiazole (Figure 2).These chemical compounds are of the azole family with the molecular formula C_2_H_2_N_2_O. Among these isomers, 1,2,3-oxadiazole is unstable and ring-opens to form the diazoketone tautomer. However, 1,3,4-oxadiazole is a thermally stable aromatic molecule and plays a major role in developing new drug candidates with diverse biological activities such as anticancer, antiparasitic, antifungal, antibacterial, antidepressant, antitubercular and antiinflammatory, etc. [13].

The electrophilic-substitution reaction is very difficult at the carbon atom in the oxadiazole ring because of the relatively low electron density on the carbon atom. However, the electrophilic attack occurs at nitrogen if the oxadiazole ring is substituted with electron releasing groups. Similarly, the oxadiazole ring is usually resistant to nucleophilic attack. However, the halogen-substituted oxadiazole undergoes nucleophilic substitution with the replacement of halogen atom by nucleophiles [14]. Although 1,3,4-oxadiazole ring system was known in 1880, significant studies were carried out regarding its chemistry, structure, physical properties and application of its various derivatives from 1950 (Table 1). 1,3,4-oxadiazole is a liquid with a boiling point of 150 °C. The percentage of C, H, N present in 1,3,4-oxadiazole is 34.29%, 2.88%, 40.00%, respectively [15].

The first monosubstituted 1,3,4-oxadiazoles were reported in 1955 by two independent laboratories. Since 1955, other research groups have performed the reactions of 1,3,4-oxadiazole and reported that it is a liquid that boils at 150°C. Ainsworth first prepared 1,3,4-oxadiazole (**2**) in 1965 by the thermolysis of ethylformate formyl hydrazone (**1**) at atmospheric pressure as depicted in Scheme 1 [16].

The 1,2,4-oxadiazole was synthesized first time in 1884 by Tiemann and Kruger. Most of the oxadiazole synthesis is based on heterocyclization of amidoxime and carboxylic acid derivatives or 1,3-dipolar cycloaddition of nitrile and nitrile oxide [17]. Microwave irradiation can also be applied in the heterocyclization of amidoximes and acyl chlorides/carboxylic acid esters in the presence of NH_4_F/Al_2_O_3_ or K_2_CO_3_ to produce corresponding oxadiazole derivatives [18]. Similarly, oxadiazole derivatives are produced by the reaction of aryl-nitrile with hydroxylamine hydrochloride to aryl-amidoxime inthe presence of a catalyst (MgO or CH_3_COOH or KF) under a microwave-assisted method. In the year 2017, Baykov et al. reported a study on the first one-pot synthetic procedure for the synthesis of 3,5-disubstituted-1,2,4-oxadiazoles (**3**) at room temperature from corresponding amidoximes (**1**) and carboxylic acids methyl or ethyl esters (**2**) in the presence of superbase medium NaOH/DMSOas presented in the Scheme 2 [19,20].

Gorjizadeh et al. reported the efficient synthesis of a series of 1,3,4-oxadiazoles (**3**) from the cyclization–oxidation reaction of acyl hydrazones (**1**) with substituted aldehydes (**2**) by using 1,4-bis(triphenylphosphonium)-2-butene peroxodisulfate (BTPPDS) as an oxidant in a solvent-free medium under microwave irradiation (Scheme 3). The reaction was found to proceed smoothly under microwave irradiation within 25 min, whereas 12 h were required to complete the reaction under reflux conditions [21].

## 4. Therapeutic Potentials of Oxadiazole Derivatives

Various medicinally active drug molecules containing oxadiazole moiety are used clinically for the treatment of different disease states (Figure 3).Oxolamine possesses a 1,2,4-oxadiazole ring and is used as a cough suppressant.Similarly, prenoxdiazine is a cough suppressant. Its IUPAC name is 3-(2,2-diphenylethyl)-5-(2-piperidin-1-ylethyl)-1,2,4-oxadiazole. Proxazole is chemically N,N-diethyl-2-[3-(1-phenylpropyl)-1,2,4-oxadiazol-5-yl]ethanamine. It is a drug used for functional gastrointestinal disorders. Butalamine is a vasodilator and is chemically known as N′,N′-dibutyl-N-(3-phenyl-1,2,4-oxadiazol-5-yl)ethane-1,2-diamine.

Ataluren (PTC124) is a drug for the treatment of Duchenne muscular dystrophy. It was designed by PTC Therapeutics and is sold under the trade name Translarna in the European Union. Chemically, it is 3-[5-(2-fluorophenyl)-1,2,4-oxadiazol-3-yl]benzoic acid. Pleconaril is an antiviral drug. It exhibits its action by inhibiting viral replication. It was developed by Schering-Plough for the prevention of asthma exacerbations and common cold symptoms in patients exposed to picornavirus respiratory infections. Chemically, it is 3-[3,5-dimethyl-4-[3-(3-methyl-1,2-oxazol-5-yl)propoxy]phenyl]-5-(trifluoromethyl)-1,2,4-oxadiazole [22,23,24].

Carbone M. et al. isolated two indole alkaloids, phidianidine-A and phidianidine-B (Figure 4), from sea slug opisthobranch *Phidiana militaris.* Chemically, phidianidine-B is 2-[5-[[5-(1H-indol-3-ylmethyl)-1,2,4-oxadiazol-3-yl]amino]pentyl]guanidine. Both phidianidines exhibit in vitro cytotoxic activity against tumor and non-tumor mammalian cell lines (rat glial-C6, human cervical-HeLa, colon adenocarcinoma-CaCo-2, mouse embryo-3T3-L1 and rat heart myoblast-H9c2) [25]. Similarly, quisqualic acid is a naturally occurring compound bearing 1,2,4-oxadiazole moiety. It is obtained from seeds of *Quisqualis indica.* It is usedfor the treatment of stroke, epilepsy and neurodegenerative disorders [26,27].

Tiodazosin is a new antihypertensive drug, structurally resembles prazosin. It possesses alpha-adrenergic-blocking activity and exerts a direct vasodilation effect. Chemically, it is (4-(4-amino-6,7-dimethoxyquinolin-2-yl)piperazin-1-yl)(5-(methylthio)-1,3,4-oxadiazol-2-yl)-methanone [28]. Furamizole is a nitrofuran derivative and possesses antibacterial activity. Chemically, it is (E)-5-(1-(furan-2-yl)-2-(5-nitrofuran-2-yl)vinyl)-1,3,4-oxadiazol-2-amine.Raltegravir (MK-0518) is an antiretroviral drug produced by Merck and Co., Kenilworth, NJ, USA. It was approved by the U.S. Food and Drug Administration (USFDA) in October 2007. It is the new class of anti-HIV drug and acts as an integrase inhibitor. Chemically, it is N-(4-fluorobenzyl)-5-hydroxy-1-methyl-2-(2-(2-methyl-1,3,4-oxadiazole-5-carboxamido)propan-2-yl)-6-oxo-1,6-dihydropyrimidine-4-carboxamide. Nesapidilisacalcium channel blocker and exerts vasodilation effect. Its IUPAC name is1-(3-(1,3,4-oxadiazol-2-yl)phenoxy)-3-(4-(3-methoxyphenyl)piperazin-1-yl)propan-2-ol (Figure 5, Table 2) [29,30].

## 5. Microwave-Assisted Synthesis of Biologically Active Oxadiazoles

Microwave-assisted drug synthesis is a Green technology that utilizes microwave radiation as a heating source to perform various synthetic reactions. The microwave radiation is used as an alternative energy source to complete various organic transformations in minutes instead of hours or even days. Microwaves are electromagnetic radiation with wavelengths ranging from one meter to one millimeter with frequencies between 300 MHz and 300 GHz [31]. These high-frequency electric fields of the microwave are applied to heat the reacting substances of an organic reaction with electric charges. In the case of the polar reaction medium, these are heated due to their dipolar rotation with the electric field and loose energy during collisions between reacting molecules. With the help of microwave heating technology, the rate of organic synthesis can be improved, and the drug products can be manufactured selectively by utilizing suitable microwave parameters. Thus, microwave technology provides several advantages such as instantaneous, rapid heating, homogeneity and selective heating as compared to conventional heating techniques like water bath, oil bath or sand batch, etc. [32].

The synthesis of drug substances under microwave irradiation is primarily dependent on the ability of the reaction medium to absorb microwave energy efficiently and also depends on the selection of the solvent system to perform the synthetic reaction [33]. Due to the diverse polar and ionic properties of different solvents, they interact differently with microwave radiation. Hence, the ability of suitable solvents or reaction medium to convert microwave energy to heat is called as loss tangent (δ). Based on the value of tan δ, solvents can be categorized into high (tanδ > 0.5), medium (tanδ 0.1–0.5), and low microwave absorbing (tanδ < 0.1) type. Higher the value of tan δ represents that the solvent is suitable for absorption of microwave radiation so as to causes efficient heating [34].

Polar solvents such as DMA, DMF, DMSO, NMP, methanol, ethanol, and acetic acid are selected for carrying out organic synthesis under microwaves due to their polarity. The solvents with low boiling points (e.g., methanol, dichloromethane and acetone) have lower reaction temperatures due to the pressure developed inside the reaction vessel. If a higher temperature is desirable to achieve a faster reaction, it is suggested to change the closely related solvent with a higher boiling point (e.g., dichloroethane instead of dichloromethane). Solvents can behave differently at elevated temperatures, and most of the solvents become less polar with an increase in temperature. For example, the bond angle in water widens at elevated temperatures, and its dielectric properties approach the organic solvents. Similarly, water at 250 °C possesses similar dielectric properties like acetonitrile at room temperature. Hence, water can be used as a pseudo-organic solvent at elevated temperatures where organic compounds will dissolve, not only because of the temperature but also because of the change in dielectric properties. Nonpolar solvents (e.g., toluene, dioxane, THF) can only be heated if other components in the reaction mixture respond to microwave energy [35,36].

However, ionic liquids (ILs) are reported as new, environmentally friendly, recyclable alternatives to dipolar aprotic solvents for organic synthesis. ILs are salts consisting of ions, which exist in the liquid state at ambient temperatures (<100 °C or 212 °F). [EtNH_3_][NO_3_] was the first example of a protic ionic liquid with a melting point of 12 °C. The ionic liquids possess unique physicochemical properties, such as negligible vapor pressure, high ionic conductivity, excellent solubility with many substances, high thermal and chemical stabilities. The dielectric properties of ionic liquids make them most suitable to utilize as solvents in microwave accelerated organic synthesis. As ionic liquids possess ions with low vapor pressure, they absorb microwave irradiation more efficiently [37,38].

Various oxadiazole derivatives are subjected to synthesis under microwave irradiation, and their biological activities are reported that include antibacterial, antifungal, antiviral, anticonvulsant, antitumor, antimalarial, antitubercular, anti-asthmatic, antidepressant, antidiabetic, antioxidant, antiparkinsonian, analgesic and antiinflammatory, etc. (Figure 6) [39,40,41,42,43,44].

Microwave-assisted synthesis permitresearcher to synthesize drug molecules with high purity and to scale up the experiments reliably from milligrams to larger quantities without altering the reaction conditions. It provides precise control over conditions of temperature and pressure as compared to conventional heating methods. In the case of a solvent-less or solvent-free reaction, the microwave energy is directly absorbed by the reactant molecules [45,46,47,48].

### 5.1. Oxadiazole Derivatives as Analgesic and Antiinflammatory Agents

Hyperalgesia refers to the enhancement in sensitivity to pain caused by nociceptors or peripheral nerves. Analgesics are the agents that relieve the pain by acting selectively on the CNS and peripheral pain mediators without changing consciousness. Analgesics may be narcotic or non-narcotic. Inflammation is a complex biological response that occurs when the body is exposed to infective agents or to physical or chemical injury. Inflammation may be three types such as acute (inflammation lasts for few days), subacute (inflammation lasts for 2–6 weeks) and chronic inflammation (inflammation lasts for months or years). The inflammatory mediators are the messenger that acts on blood vessels or cells to promote the inflammatory responses. Various inflammatory mediators include prostaglandins (PGs), inflammatory cytokines such as IL-1β, TNF-α, IL-6 and IL-15 and chemokines such as IL-8 and GRO-alpha. The treatment of inflammation depends on the cause and severity. Due to the presence of common side effects of NSAIDs (gas, bloating, heartburn, ulcer, stomach pain), it is required to develop new drugs for the treatment of pain and inflammation without any side effects. Hence, theoxadiazole derivatives play a major role in the treatment of pain and inflammation as compared to NSAIDs [49,50].

Biju et al. reported the design and microwave-assisted synthesis of 1,3,4-oxadiazole derivatives for analgesic and antiinflammatoryactivity. In step-I, a mixture of isoniazid, aromatic aldehyde and DMF (5 drops) was subjected to microwave irradiation at 300 w for 3 min to get the solid product (**1a**). In step-II, chloramine-T was added to the solution of compound 1a in ethanol. The reaction mixture was subjected to microwave irradiation at 300W for 4 min to yield 2-aryl-5-(4-pyridyl)-1,3,4-oxadiazole (**2a–i**) (Scheme 4). Among the newly synthesized 1,3,4-oxadiazole analogs, compounds 2a, 2c and 2i displayed good analgesic and antiinflammatory activity as compared to the standard drug, indomethacin. From the acute toxicity study, it is revealed that the oxadiazole analogs are safe with low toxicity [51].

Frank et al. reported the microwave-assisted as well as the conventional synthesis of 5-substituted-2-(2-methyl-4-nitro-1-imidazomethyl)-1,3,4-oxadiazoles containing the nitroimidazole moiety (Scheme 5).These compounds are evaluated forantiinflammatory activity based on the method of Winter et al. Formalin-induced edema test was employed. All the tested compounds exhibitantiinflammatory activity at a dose of 50 mg/kg. The antiinflammatory activity of **7b**, **7c** and **7d** is compared with that of a standard drug (indomethacin) at a dose of 1⋅5 mg/kg [52].

Sahoo et al. reported a series of various Schiff bases of 1,3,4-oxadiazole analogs (Scheme 6). These compounds are synthesized by utilizing the principles involved in green chemistry that significantly reduce the chemical waste and reaction time. To demonstrate these advantages in the synthesis of bioactive oxadiazole derivatives, various environmentally benign protocols that involve greener alternatives are studied. The efficiency of microwave heating technology has resulted in remarkable reductions of reaction times from hours to minutes with better product yield. The antiinflammatoryactivity of the selected compounds is evaluated. This result indicates that some of the compounds exhibit significant activity as compared to standard Indomethacin [53].

Both conventional and microwave-assisted methods are applied to synthesize various 2-(4′-methylbiphenyl-2-yl)-5-aryl-1,3,4-oxadiazole analogs (**4a–n**) (Scheme 7). These compounds are further evaluated for analgesic and antiinflammatory activity by invivomodels. The tested compounds exhibited significant analgesic and antiinflammatory activities as compared to standard drugs. Docking studies are performed to determine the binding mode of designed compounds with the COX-2 enzyme. The pharmacokinetic parameters are calculated for the synthesized compounds and are found to be within the acceptable range defined for human use revealing their potential as possible drug-like compounds. Hence, the results obtained indicate that these compounds can serve as good leads for further modification and optimization [54].

### 5.2. Oxadiazole Derivatives as Antioxidant and Antimicrobial Agents

Farshori et al. reported the facile one-pot synthesis of novel 2,5-disubstituted-1,3,4-oxadiazoles under conventional and microwave conditions (Scheme 8) and evaluation of their in vitro antimicrobial activities. The newly synthesized compounds are screened for their antibacterial activity against *Escherichia coli* (ATCC-25922), methicillin-resistant *Staphylococcus aureus* (MRSA +Ve), *Pseudomonas aeruginosa* (ATCC-27853), *Streptococcus pyogenes* and *Klebsiella pneumoniae* (clinical isolate) bacterial strains by disc diffusion method. Ciprofloxacin (30 µg) is used as a positive control.

Similarly, antifungal activity is determined by the disk diffusion method against *Candida albicans*, *Aspergillus fumigatus*, *Penicillium marneffei* and *Trichophytonmentagrophytes (recultured)* in DMSO. The fungal activity of each compound is compared with greseofulvin as a standard drug. The compounds **3f**, **3j** and **3l**, are found to be the most potent antimicrobial agents. The lowest concentration (highest dilution) required to arrest the growth of the fungus is regarded as a minimum inhibitory concentration (MIC). Similarly, MFC is defined as the lowest drug concentration at which 99.9% of the inocula are killed [55].

A new series of novel chromene-based oxadiazole derivatives are synthesized from a variety of Chromene-based amidoximes with readily available carboxylic acids under both conventional heating and microwave irradiation method (Scheme 9). The coupling reagents such as EDCI and HOBt in DMF are subjected to microwave heating that results in high yields and purities of the product, 1,2,4-oxadiazoles in an expeditious manner. The results were obtained for both conventional and microwave irradiation methods. It indicates that microwave irradiation led to an enhancement in the rate as well as the yield of the products (80–92%) over the conventional method (65–78%). All the synthesized compounds are evaluated for their invitro antibacterial activity against two different pathogenic bacterial strains, such as *Escherichia coli* (MTCC614) and *Klebsiella pneumoniae* (MTCC4031). The obtained results reveal that 6g and 6h exhibited good antibacterial activity as compared to the standard drug (Gentamicin) [56].

Bodke et al. reported the conventional and microwave-assisted synthesis of 5-phenyl-2-substituted-1,3,4-oxadiazole derivatives (Scheme 10). The title compounds are screened for their antimicrobial and antioxidant activity. The antimicrobial activity of the synthesized compounds is performed against four bacterial strains and two fungal strains (*Chrysosporium keratinophilum* and *Candida albicans*) by using the agar well diffusion method. The compounds **3e** and **3g** are endowed with high antibacterial activity when compared with standard antibacterial drugs. Among all tested compounds, **3d** and **3n** displayed significant activity against both fungal pathogens with MIC values 750 μg/mL. Similarly, all the synthesized compounds are screened for free radical scavenging activity by the DPPH method. Among the tested compounds, **3a**, **3b**, **3c**, **3e**, and **3f** have potential radical scavengingactivity, while compounds **3a**, **3b**, **3c**, **3h**, and **3j** have the potential metal chelating ability as compared to the standard drug. The antioxidant results revealed that the compounds exhibit antioxidant activity in a dose-dependent manner [57].

### 5.3. Oxadiazole Derivatives as Antimycobacterial Agents

A series of newer analogs of 5-(pyridine-4-yl)-1,3,4-oxadiazole-2(3H)-thione is designed and synthesized by incorporating 1,3,4-oxadiazole and bezo[d]thiazole by Mannich base reaction using conventional as well as microwave irradiation method (Scheme 11). This process has considerable advantages such as milder reaction conditions, an accelerated rate of reaction, less time-consuming and higher product yields as compared to conventional heating methods.

The synthesized compounds are evaluated for their antimycobacterial activity against *M. tuberculosis*. Among the tested compounds, compound 1c containing methyl group at ortho position on an aromatic ring exhibited better activity against *M. tuberculosis* H37Ra using the MABA method. Hence, the modification of substituents on bezo[d]thiazole ring with various electron releasing and electron-withdrawing groups affect the activity [58].

### 5.4. Oxadiazole Derivatives with Antitumor Activity

Oliveira et al. focused on microwave irradiation-assisted synthesis of N-cyclohexyl-1,2,4-oxadiazole derivatives with antitumor activity. Arylamidoximes (**1a–i**) and dicyclohexyl-carbodiimide (DCC) in DMF is subjected to focused microwave irradiation (FMWI) to obtain 1,2,4-oxadiazoles (**2a–i**) with 61–81% yields (Scheme 12). All these compounds exhibit antiproliferative activity invitro against three human cancer cell lines, HCT-116, PC-3, and SNB-19. The cytotoxicity of the compounds is evaluated by using the MTT assay method.Compounds such as **2b** (m-CH_3_) and **2c** (m-Br) are found to have antitumor activity againstSNB-19 (13.62 mM) and PC-3 (21.74 mM), respectively [59].

Modi et al. reported the synthesis of novel achiral and chiral amides incorporating the 1,3,4-oxadiazole ring by applying both conventional and microwave methods (Scheme 13). The synthesis of these compounds by microwave method produces comparatively more yield and requires less time to complete the reaction.The synthesized compounds are screened for microbial and cytotoxic activities. The synthesized compounds are tested for their antibacterial activity *invitro* against two microorganisms viz. *Escherichia coli* and *Staphylococcus aureus*, which are pathogenic in human beings by cup plate agar diffusion method. The tested compounds showed moderate to good microbial activities.

Similarly, these compounds are tested for their antifungal activity *invitro* against *Aspergillus oryzae* and *Aspergillus niger* by cup-plate agar diffusion method. Further, all the synthesized compounds (**Ia–h**) are tested for cytotoxic activity by the BSLT bioassay method. Among them, compounds Ia, Ic, Ih exhibited cytotoxic activity in a dose-dependent manner at concentrations of 24.27 µg/mL, 37.05 µg/mL, 39.26 µg/mL, respectively. Podophyllotoxin is used as a standard drug for the BSLT assay method [60].

### 5.5. Miscellaneous

Saravanan et al. reported the facile synthesis of N-1,2,4-oxadiazole substituted sulfoximines from N-cyano sulfoximines (Scheme 14). This method involves the utilization ofa wide range of substrate thatincludes alkyl, fluoroalkyl, aryl, heteroaryl and saturated heterocyclic compounds. The major advantage of this method is metal-free and the utility of environmentally friendly solvents such as 2-methyl THF and ionic liquids [61].

Bharatiya et al. reported the green and efficient microwave one-pot synthetic approachto N-phenyl piperazinyl-1,3,4-oxadiazole derivatives (Scheme 15) and evaluation of their antioxidant and antiinflammatory activity. Microwave-assisted synthesis has proved to be a green and efficient synthetic protocol to enhance reaction rates, higher selectivity, efficiency and higher yield. The one-pot synthetic strategy includes successive reactions in just one reaction vessel to improve the efficiency of a chemical reaction. This synthetic protocol avoids the lengthy separation process and purification of the intermediates and hence increases the product yield.The synthesized compounds were screened for invitroantiinflammatory activity. The antioxidant activity of the synthesized compounds was determined by reducing power assay and hydrogen peroxide scavenging activity at 700 nm and 250 nm, respectively.The tested compounds exhibited significant antiinflammatory and antioxidant activity [62].

Sondhi et al. performed the microwave-assisted synthesis of oxadiazole derivatives (Scheme 16) with antiinflammatory and anticancer activities. N’-hydroxypicolinamidine and 2,5-dimethoxybenzaldehyde were mixed together to furnish a homogeneous mixture, and then this reaction mixture was subjected to microwave irradiation for 3 min at a power level of 450 W.Antiinflammatory activity (Winter et al.) evaluation of IIa-c and IVa–f was carried out by using carrageenan-induced paw edema assay.The result indicates that compounds IIj, IIk, and IVb exhibited 35%, 34%, and 35% antiinflammatory activity, respectively, whereas standard drug ibuprofen exhibited 39%antiinflammatory activity at 50 mg/kg p.o. Similarly, compound IVd exhibited good anticancer activity against various cancer cell lines, i.e., lung (NCIH-522), breast (T47D), liver (HepG2) [63].

A rational approach was adopted by Desai et al.for the synthesisof novel series of 1-(2-(1H-benzo[d]imidazol-2-yl)-2-methyl-5-aryl-1,3,4-oxadiazol-3(2H)-yl)-3-(4-chlorophenyl)prop-2-en-1-ones (**5a–n**) under microwave irradiation technique (Scheme 17). The synthesized compounds were screened for their in vitro antimicrobial activity against Gram-positive, Gram-negative strains of bacteria as well as fungal strains, and in vitro cytotoxicity study (HeLa cell lines) by using MTT colorimetric assay. Among the tested compounds, **5b**, **5c**, **5d**, **5g**, and **5h** exhibited the most potent antibacterial activity, while compounds **5c**, **5d**, **5g**, and **5h** emerged as the most effective antifungal agents. Further, the results of a preliminary MTT cytotoxicity study on HeLa cells revealed that potent antimicrobial activity of **5b**, **5c**, **5d**, **5g**, and **5h** is accompanied by low cytotoxicity. The structure–activity relationships (SAR) study suggested that the presence of inductively electron-withdrawing groups (fluoro, chloro, bromo, nitro, methoxy, hydroxy) remarkably enhances the activity of newly synthesized oxadiazole derivatives [64].

A series of 1,3,4-oxadiazole derivatives were designed, synthesized and evaluated for their radical scavenging and antiinflammatory activity (Scheme 18). Molecular docking simulation studies were performed on the proteins cyclooxygenase-1 (PDB: 1CQE) and cyclooxygenase-2 (PDB: 3LN1) to visualize the probable binding affinity.This insilico study was used to predictthe appreciable ADME and probable toxicity property of the compounds. The best-ranked 1,3,4-derivatives (**5a–5j**) were synthesized from 2-(benzo[d]thiazol-2-ylamino)acetohydrazide (**4**) on reaction with aryl/heteroaryl/aliphatic carboxylic acid derivatives via acid-catalyzed dehydrative cyclization. N-((5-mercapto-1,3,4-oxadiazol-2-yl)methyl)benzo[d]thiazol-2-amine (**5k**) was synthesized by base-catalyzed condensation of hydrazide derivative 4 and with carbon disulfide. The 1,3,4-oxadiazoles were evaluated for in vitro antioxidant activity by 2,2′-diphenyl-1-picryl hydrazyl (DPPH) radical scavenging assay method and in vivo antiinflammatory activity by carrageenan-induced paw edema method. The results of radical scavenging activity indicate that the 1,3,4-oxadiazoles at 25 µM test concentration exhibited significant radical scavenging property ranging from 32.0 to 87.3% in comparison to 76.0% radical scavenging activity obtained for the reference drug (ascorbic acid). The results of the in vivo antiinflammatory activity suggested that the 1,3,4-oxadiazoles at 25 mg kg^−1^ test dose exhibited significant edema inhibition with a mean value ranging from 23.6 to 82.3% in comparison to 48.3% edema inhibition obtained for the reference drug (Indomethacin) [65].

## 6. Ultrasound-Mediated Synthesis of Oxadiazole Derivatives

Ultrasound-mediated organic synthesis is a green synthetic approach and applied as a powerful technique to enhance the rate of reaction and product yield. The ultrasonic irradiation is enhanced due to the formation of high energy intermediates.This synthetic method can be considered as an eco-friendly process for the conservation of energy and minimization of waste as compared to the conventional techniques. Similarly, molecular sieves assisted synthesis has attracted considerable attention due to their potential use in catalysis [66].

Nikalje et al. reported the ultrasound and molecular sieves assisted synthesis, molecular docking and antifungal evaluation of 5-(4-(benzyloxy)-substituted-phenyl)-3-((phenylamino)methyl)-1,3,4-oxadiazole-2(3H)-thiones. These oxadiazole derivatives were synthesized successfully by using molecular sieves under ultrasound irradiation with better product yields of 78–90% in shorter reaction times as compared to conventional heating methods, which require 15–20 h for refluxing. The titled compounds exhibited promising antifungal activity, and the developed scaffold provides a suitable template for generating antifungal agents [67].

## 7. Structure–Activity Relationship (SAR) Study

The presence of oxadiazole moiety is required for producing biological activities like antibacterial, antifungal, anticancer, antitubercular, antioxidant, analgesic and antiinflammatory, etc. [68].

Position C_2_ and C_5_ are essential for substitutions to generate different oxadiazole derivatives.

The presence of electron-withdrawing substituents (fluoro, chloro, bromo, nitro, methoxy and hydroxyl) potentiates the activity by imparting high-affinity and selectivity towards the target binding site of the receptor [69].

The intense biological activity of the compounds is also greatly influenced by the amount of activation or deactivation and the position of the groups or substituents on the ring.

The presence of aryl or hetero aryl group in the side chain of the oxadiazole nucleus is required for the lipophilic property of drug molecules.

Structural modification of the oxadiazole scaffold can generate different therapeutically active compounds (Figure 7) [70].

## 8. Future Development

Microwave radiation acts as a nonconventional energy source that can be used to perform a wide range of drug synthesis within a short period of time with high yields as compared to conventional heating methods.The chemical reactions which are not possible under conventional techniques can sometimes be carried out by utilizing the high energy of MWI. By applying microwave technology, different oxadiazole derivatives can be synthesized and also screened to find out new therapeutic molecules with diverse biological activities such as antibacterial, antifungal, antidepressant, antitubercular and antiinflammatory, etc. [71,72,73,74]. Hence, oxadiazole is considered a significant heterocyclic core and becomes a major scaffold for the development of new drug candidates because of its potential to be involved in the binding interactions with different targets or receptors with suitable metabolic profile [75,76,77,78].

## 9. Conclusions

The synthesis of various heterocyclic compounds like oxadiazole derivatives under microwave irradiation demonstrates several advantages in terms of remarkably short reaction time, high product yields and a simple purification process in comparison with the classical synthetic strategies.Moreover, also, the volume of solvents used during a chemical reaction is reduced, which makes MWI an environmentally friendly synthetic approach. Further, oxadiazole derivatives have attracted medicinal chemists in search of new therapeutic agents due to the presence of diverse molecular structures. The therapeutic potentials of oxadiazole rings have been extensively studied to develop selective drug molecules inwhich the presence of different substituents or groups in the molecule is responsible for displayingdifferent pharmacologicalactivities. In this review, the synthesisof oxadiazole derivatives by the conventional method and themicrowave method is reported to compare their effective synthetic strategies followed by a study of theirdifferent therapeutic activities. From various results, it reveals that the structural modification or functionalization of oxadiazole scaffoldsto generate a compound library with diverse biological activities.

## Data Availability

Not applicable.

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
