# Peer review of "Green Synthetic Approach: An Efficient Eco-Friendly Tool for Synthesis of Biologically Active Oxadiazole Derivatives"

_molecules, 2021, doi:10.3390/molecules26041163_

Round 1

Reviewer 1 Report

This manuscript entitled: Green Synthetic Approach: An Efficient Eco-Friendly Tool for Synthesis of Biologically Active Oxadiazole Derivatives describes synthesis and biological activity of oxadiazol derivatives. This manuscript gives the impression of a well-crafted review. The chapters are logically arranged and the description corresponds to common standards. But there are a number of fundamental shortcomings in the detailed analysis. I see a fundamental problem in the lack of an introduction section. The introduction explains the concept of green chemistry and follows a very brief introduction to the chemistry of oxadiazoles. Unfortunately, this section lacks a definition of the scope of review in the context of other works. It is remarkable that the authors ignored the work of their colleagues and did not cite the recent review. SciFindern has found more than 600 reviews on oxadiazoles, see selected examples:

Anti-tuberculosis activity and its structure-activity relationship (SAR) studies of oxadiazole derivatives: A key review

By: Verma, Santosh Kumar; Verma, Rameshwari; Verma, Shekhar; Vaishnav, Yogesh; Tiwari, S. P.; Rakesh, K. P.

European Journal of Medicinal Chemistry (2021), 209, 112886

Antimalarial activity of oxadiazoles - a review

By: Hasna, K. T.; Deepika, P.; Sherin, A.; Shiji, Kumar P. S.

International Journal of Pharmaceutical Sciences Review and Research (2020), 61(1), 89-92

Antiviral activity of thiadiazoles, oxadiazoles, triazoles and thiazoles

By: Tawfik, Samar S.; Liu, Mengyao; Farahat, Abdelbasset A.

ARKIVOC (Gainesville, FL, United States) (2020), (1), 180-218

Novel 1,2,4-oxadiazole derivatives in drug discovery

By: Biernacki, Karol; Dasko, Mateusz; Ciupak, Olga; Kubinski, Konrad; Rachon, Janusz; Demkowicz, Sebastian

Pharmaceuticals (2020), 13(6), 111

Groundbreaking Anticancer Activity of Highly Diversified Oxadiazole Scaffolds.

By: Benassi, Alessandra; Doria, Filippo; Pirota, Valentina

International journal of molecular sciences (2020), 21(22)

Without a proper determination of the scope of the manuscript, the work cannot be recommended for publication in molecules. In addition, from a formal point of view, a number of shortcomings need to be addressed:

The logical numbering of substances in the schemes is missing.

Substance names should not be part of the schemes (eg Scheme 1, 4, 7).

Some structures are ugly (I mean bond angles, atom alignment etc): eg. nesapidil and furamizole (Figure 5), compound (I) (Scheme 15), compounds 5a-n (Scheme 17)

Schemes 16, 17, 18 have low dpi. Please replace these schemes with better improves ones

some abbreviations are unnecessary: °C, mg/kg, mM, W and others are SI units and not abbreviations. The same applies to chemical names: acetic acid, magnesium oxide, sodium hydroxide, ammonium fluoride, aluminium oxide.

Moreover, references should also be checked because the style of writing references is not consistent.

In view of these findings I would recommend to proofreading the manuscript by a professional.

Reviewer 2 Report

This review paper by Banik and coauthors summarized the progress for the synthetic approach toward oxadiazole derivatives. This work is potentially interesing to researchers working on drug synthesis. The paper is overall well-written. The authors need to pay more attention to their chemdraw structure:

  1. In scheme 4 and scheme 2, bond angle around a sp2 carbon should be 120 degrees.
  2. Composition of functional group should be written carefully. In Scheme 7, H2NHNOC should be draw as bonds and atoms, or written unambigously.
  3. Units for temperature. Scheme 14, the temperature symbol.
  4. Size of atom labels and lengths of bonds should be consistent through every scheme within this paper. In sheme 15, the atom labels are too small to be easily readable compared to other schemes.

Round 2

Reviewer 1 Report

Although the manuscript still requires further editing from a formal point of view, I recommend publishing in Molecules.

This manuscript is a resubmission of an earlier submission. The following is a list of the peer review reports and author responses from that submission.